# The cross-sectional association of stressful life events with depression severity among patients with hypertension and diabetes in Malawi

Kelsey R. Landrum[1]*, Brian W. Pence[1], Bradley N. Gaynes[1,2], Josée M. Dussault[1], Mina C. Hosseinipour[3,4], Kazione Kulisewa[5], Jullita Kenela Malava[4], Jones Masiye[6], Harriet Akello[4], Michael Udedi[6], Chifundo C. Zimba[4]

1 University of North Carolina at Chapel Hill, Department of Epidemiology, Gillings School of Global Public Health, Chapel Hill, North Carolina, United States of America, 2 University of North Carolina at Chapel Hill, Department of Psychiatry, Chapel Hill, North Carolina, United States of America, 3 University of North Carolina at Chapel Hill, Department of Medicine, Chapel Hill, North Carolina, United States of America, 4 UNC Project Malawi, UNC Project, Tidziwe Centre, Lilongwe, Malawi, 5 Kamuzu University of Health, Department of Psychiatry and Mental Health, Blantyre, Malawi, 6 Malawi Ministry of Health, Noncommunicable Diseases and Mental Health Unit, Lilongwe, Malawi

* klandrum@email.unc.edu

**Data Availability Statement:** De-identified, individual-level SHARP data is shared via the National Institute of Mental Health Data Archive

## Abstract

Depressive disorders are a leading cause of global morbidity and remain disproportionately high in low- and middle-income settings. Stressful life events (SLEs) are known risk factors for depressive episodes and worsened depressive severity, yet are under-researched in comparison to other depression risk factors. As depression is often comorbid with hypertension, diabetes, and other noncommunicable diseases (NCDs), research into this relationship among patients with NCDs is particularly relevant to increasing opportunities for integrated depression and NCD care. This study aims to estimate the cross-sectional association between SLEs in the three months preceding baseline interviews and baseline depressive severity among patients with at least mild depressive symptoms who are seeking NCD care at 10 NCD clinics across Malawi. SLEs were measured by the Life Events Survey and depressive severity (mild vs. moderate to severe) was measured by the Patient Health Questionnaire-9. The study population (n = 708) was predominately currently employed, grand multiparous (5–8 children) women with a primary education level. Two thirds (63%) had mild depression while 26%, 8%, and 3% had moderate, moderately severe, and severe depression, respectively. Nearly all participants (94%) reported at least one recent SLE, with the most common reported SLEs being financial stress (48%), relationship changes (45%), death of a family member or friend (41%), or serious illness of a family member or friend (39%). Divorce/separation, estrangement from a family member, losing source of income, and major new health problems were significant predictors of greater (moderate or severe) depressive severity compared to mild severity. Having a major new health problem or experiencing divorce/separation resulted in particularly high risk of more severe depression. After adjustment, each additional SLE was associated with a 9% increased risk of moderate or worse depressive severity compared to mild depressive

(https://ndar.nih.gov/contribute_data_sharing_regimen.html) and is updated every 6 months. Researchers can apply to the National Institute of Mental Health Data Archive to access these data. The data used and analyzed in this study are also available from the corresponding author and the parent study principal investigator upon a reasonable request.

**Funding:** This study received funding from the National Institute of Mental Health (MPIs: Brian W. Pence, PhD, Mina C. Hosseinipour, MD MPH; Jones Masiye; U19MH113202-01). KRL received support from supplement U19MH113202-04S1. JD received support from supplement U19MH113202-01S2 and T32AI070114. Neither the funding agency nor funders had any part in study conceptualization, data collection, data analysis, or decision-making about publication.

**Competing interests:** The authors have declared that no competing interests exist.

**Abbreviations:** DAG, Directed Acyclic Graph; DALY, Disability adjusted life years; GAD, Generalized Anxiety Disorder; LMICs, Low- and middle-income countries; MNS, Mental, neurological, and substance abuse; NCD, Non-communicable disease; PHQ-9, Patient Health Questionnaire-9; PR, Prevalence ratio; PTSD, Post-traumatic stress disorder; RCT, Randomized control trial; SHARP, Sub-Saharan Africa Regional Partnership for Mental Health Capacity Building; SLE, Stressful life event; YLD, Years lived with disability.

severity (RR: 1.09; (95% CI: 1.05, 1.13), p<0.0001). Among patients with NCDs with at least mild depressive symptoms, SLEs in the prior 3 months were associated with greater depressive severity. While many SLEs may not be preventable, this research suggests that assessment of SLEs and teaching of positive coping strategies when experiencing SLEs may play an important role in integrated NCD and depression treatment models.

## Introduction

Depression is the leading cause of mental health-related illness globally, and a leading contributor of Disability Adjusted Life Years (DALYs) and Years Lived with Disability (YLD) among mental, neurological, and substance use (MNS) disorders [1, 2]. While not all individuals with non-communicable diseases (NCDs) develop depression, having an illness may increase one's risk if depression [3–8]. Increasing global depression rates and association of depression with hypertension, diabetes, and other NCDs highlight the importance of researching and implementing cost-effective methods of integrating depression care into existing NCD treatment interventions and care infrastructure [3–8].

Furthermore, depression prevalence remains high in many low- and middle-income countries (LMICs) despite cost-effective treatment options [9, 10]. Estimated depression prevalence in Sub-Saharan Africa ranges from 4% to 21% and depression prevalence in Malawi, the context for this study, is best estimated to be between 4–31% [11–16]. Such a range in prevalence may be related differences in depression screening tools, study populations, outcome ascertainment timelines, and study settings; however, these are best and plausible estimates given current literature and knowledge. Even so, predictors of depression and possible intervention points in populations with high NCD prevalence remain poorly understood despite recognized need to integrate mental health and NCD care and known adverse economic impacts of NCDs at micro and macro levels, particularly in low-resource settings [17–25]. Improved efficiency in directing limited resources to high-risk populations at specific intervention points could improve mental health and NCD treatment integration.

Stressful life events (SLEs) are possible risk factors for identifying those at risk of depressive episodes and worsening depressive severity. SLEs are acute or chronic events that require adjustment of usual activities and include known risk factors for depressive disorders, such as living in poverty, food insecurity, financial stress, and intimate partner violence (IPV) [26–33]. Stress sensitization hypotheses posit that after experiencing severe SLEs, less severe SLEs can trigger repeated depressive episodes more easily [34–36]. However, research on the association between SLEs of various severities and depression severity remains lacking in low-resource and NCD settings [37].

Our goal was to examine the cross-sectional association of recent SLEs with heightened current depressive severity in a cohort of patients with NCDs with mild to severe depression in Malawi [38]. We hypothesized that participants who report exposure to more stressful life events in the 3-month period before baseline interviews would present with more severe depressive symptoms at baseline.

## Methods

### Sample and procedure

This analysis uses baseline data from a cohort of patients identified with mild, moderate, moderately severe, or severe depressive severity who were receiving medical care a participating hospital based NCD clinic (n = 10) across Malawi from May 9, 2019 until November 9, 2021.

The 10 recruitment clinics were participating in a cluster-randomized trial comparing implementation strategies to achieve integration of depression screening and treatment into NCD clinical care (the SHARP Study, NCT03711786). Patients at each clinic received that clinic's standard of care, supported by its assigned implementation strategies.

Participants were recruited if they were receiving care at a participating NCD clinic and met eligibility criteria. Eligibility criteria were defined as being aged 18–65 years, being a current or new patient receiving hypertension or diabetes care from a participating NCD clinic, and having at least mild depression during routine clinic screening (defined as a clinician-administered PHQ-9 score ≥5) [39, 40]. Patients with a history of bipolar or psychotic disorders or who showed emergent threat of self-harm were excluded. Eligible patients who provided written informed consent were enrolled in the observational cohort. After enrollment, study participants completed structured research interviews at baseline and three, six, and 12 months. This study analysis uses data from baseline research interviews. Importantly, data collection occurred before and during the SARS-CoV-2 pandemic. All data from March 25, 2020 until April 3, 2021 were collected via telephone interviews due to the SARS-CoV-2 pandemic.

## Measures

**Depression.** All participants (Table 1) completed the Patient Health Questionnaire-9 (PHQ-9) with a health care professional at their facility as part of routine clinical care, with PHQ-9 score determining their eligibility for enrollment into the study [39, 41]. Enrolled participants completed the PHQ-9 again with a trained research assistant as part of the study baseline interview, either the same day or within one week of enrollment. For analysis, participants were classified as having mild, moderate, or severe depressive symptoms at baseline based on the research assistant-administered baseline PHQ-9 score. The PHQ-9 Questionnaire (SF 1) consists of nine questions assessing the frequency of nine symptoms of major depressive disorder over the past 2 weeks as defined by the Diagnostic and Statistical Manual of Mental Disorders, 5[th] Edition [42]. Mild depression was defined as a PHQ-9 score from 5–9, inclusive [41]. Moderate, moderately severe, and severe depression were defined as a PHQ-9 score 10–14, 15–19, and 20–27, respectively [41]. For this analysis, the PHQ-9 score was dichotomized to compare mild versus moderate to severe depression.

**Stressful life events.** SLEs were defined as acute or chronic stressful events in the three months preceding the baseline interview. Participants were asked a structured series of 13 SLE questions (SF 2) during baseline interviews, adapted for use in this context from the Life Experiences Survey [43–45]. Participants were asked about relationship stressors (marriage or engagement, having an increase in serious arguments with a partner, divorce or separation, or change in closeness with a family member), death of a family member or friend, serious illness of a family member or friend, work-related events (inability to find work, losing source of income, and major problems at work), new illness or injury, hospitalization, motor-vehicle accident, experience with assault, attack, or robbery, or financial stress (represented by number of reported days going to bed hungry in the past month) [43–46].

**Covariates.** Data were collected on known and plausible confounding and modifying variables identified *a priori* with a directed acyclic graph (DAG) (Fig 1) [47]. Covariates included age, sex, and education level [48]. Covariates were included in the final model based on DAG theory and plausibility of effect on risk of SLEs and risk of more severe depression.

## Statistical analysis

We present descriptive statistics of SLE frequency and depressive severity at baseline. We fit a multivariate regression model of the association between recent SLEs and depression severity

**Table 1. Participant characteristics at baseline.**

| | | N (%) | Mean | SD | Min | Max |
|---|---|---|---|---|---|---|
| | | 708 | | | | |
| **Gender** | Female | 562 (79.4) | | | | |
| | Male | 146 (20.6) | | | | |
| | Missing | 0 (0.0) | | | | |
| **Age** | | | 60.0 | 9.8 | 18 | 65 |
| | 18–29 | 20 (2.8) | | | | |
| | 30–39 | 80 (11.3) | | | | |
| | 40–49 | 186 (26.3) | | | | |
| | 50–59 | 259 (36.6) | | | | |
| | 60+ | 163 (23.0) | | | | |
| | Missing | 0 (0.0) | | | | |
| **Education** | | | | | | |
| | No formal schooling | 112 (15.8) | | | | |
| | Standard 1–5 | 231 (32.6) | | | | |
| | Standard 6–8 | 208 (29.4) | | | | |
| | Secondary school | 120 (17.0) | | | | |
| | Post-secondary school | 37 (5.2) | | | | |
| | Missing | 0 (0.0) | | | | |
| **Marital status** | | | | | | |
| | Never married | 11 (1.6) | | | | |
| | Currently married | 467 (66.0) | | | | |
| | Separated | 66 (9.3) | | | | |
| | Divorced | 41 (5.8) | | | | |
| | Widowed | 119 (16.9) | | | | |
| | Cohabiting with a partner | 3 (0.4) | | | | |
| | Missing | 1 (0.1) | | | | |
| **Parity*** | | | 5.8 | 2.8 | 0 | 14 |
| | 0 children | 14 (2.5) | | | | |
| | 1–4 children | 171 (30.4) | | | | |
| | 5–8 children | 278 (49.5) | | | | |
| | 9–14 children | 94 (16.7) | | | | |
| | Missing | 5 (0.9) | | | | |
| **Clinic** | BW | 56 (7.9) | | | | |
| | CH | 58 (8.2) | | | | |
| | KA | 47 (6.6) | | | | |
| | KU | 69 (9.8) | | | | |
| | MA | 64 (9.0) | | | | |
| | MC | 80 (11.3) | | | | |
| | MJ | 90 (12.7) | | | | |
| | PE | 72 (10.2) | | | | |
| | SA | 102 (14.4) | | | | |
| | ZA | 70 (9.9) | | | | |
| **Work status** | | | | | | |
| | Employed | 671 (94.8) | | | | |
| | Unemployed | 33 (4.7) | | | | |
| | Missing | 4 (0.6) | | | | |

*Among female participants only

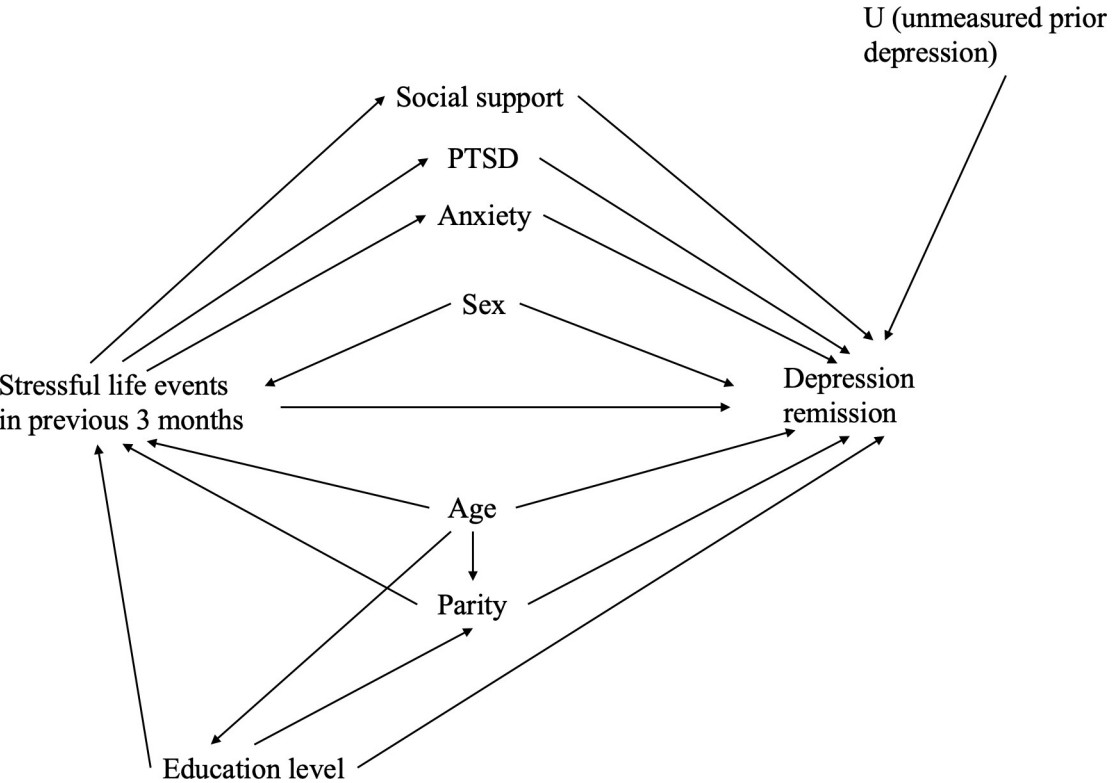

**Fig 1. Directed acyclic graph: Stressful life events and depression severity.**

at baseline. We used a modified Poisson form of the generalized linear model with robust standard errors to obtain parameter estimates interpretable, after exponentiation, as prevalence ratios (PRs) [49]. Depressive severity, the dependent variable, was coded as a binary variable representing moderate to severe (PHQ-9 score $\geq$10) vs. mild (score 5–9) symptoms, following standard PHQ-9 scoring [39, 41, 50]. The total number of SLEs per participant, the primary independent variable, was summed and coded as a continuous variable. Financial stress was represented by a proxy, categorical covariate based on reported number of days having gone to bed hungry in the past month. The appropriateness of assuming a linear relationship between number of SLEs and (log) prevalence of moderate-to-severe depressive symptoms was assessed through visualization by LOWESS plots and by a likelihood ratio test comparing a model with a quadratic exposure term to a model with only the linear term. We also considered categorical coding, comparing Akaike Information Criterion (AIC) estimates with the linear model to assess model fit.

Confounders included in the final adjustment set were identified *a priori* through Directed Acyclic Graphs (DAGs). Covariate functional forms were compared using LRTs (alpha = 0.05) and AICs for nested models and AICs for non-nested models. Alignment with our DAG theory was our primary criterion for choosing covariates. Covariates missing substantial data (>20% of observations) were not included in the final model (e.g., including parity in the model would restrict analysis to only female participants). Participants with PHQ-9 scores greater than 5 at clinician-administered study screening but less than 5 at research assistant-administered study baseline were excluded from analysis (n = 193).

### Ethical approval

This research was approved by the National Health Sciences Research Committee of Malawi (NHSRC; Approval # 17–3110) and the University of North Carolina at Chapel Hill Biomedical Institutional Review Board (Approval #17/11/1925). All participants gave written informed consent (provided in Chichewa and Chitumbuka) prior to participation in any study procedures. Participants also completed an informed consent comprehension checklist prior to providing written consent. All informed consent procedures were approved by both institutional review boards.

## Results

### Participant characteristics

Participants (n = 708, Table 1) across the 10 study sites were predominately females (n = 562, 79%) aged 50 years or greater (n = 422, 60%). Most participants reported having a primary education level grades 1–5 (n = 231, 33%) or grades 6–8 (n = 208, 29%), while 16% reported no formal schooling (n = 112) and 22% reported at least some secondary education (n = 157). Most participants were married (n = 467, 66%). Among female participants, most were grand multiparous (n = 278, 50%, range: 0–14, mean: 5.8). Most participants reported current employment (n = 671, 95%). Participant characteristics and enrollment were similar across study sites, with MC having a slightly higher proportion of male participants; MC, CH, and KA having a smaller proportion of participants with no formal education; and KA, MA, and MC having no unemployment reported at baseline.

### Depression frequency at baseline

The mean PHQ-9 score (Table 2, Fig 2) was 9.4 (SD: 3.9, range: 5–25), which would be considered on the threshold between mild and moderate depression in an individual patient. Over half of participants (n = 448, 63%) had mild depressive symptoms while n = 256 (36%) had at least moderate depressive symptoms. One quarter of participants (n = 181, 26%) had moderate depressive symptoms at baseline. More severe symptoms were less common, with 8% (n = 56) and 3% (n = 19) of participants having moderately severe and severe depressive symptoms at baseline, respectively. Of those participants who reported thoughts of self-harm (n = 128, 18%), most reported thoughts 1–7 days in the last 2 weeks (n = 105, 82%) while 8 (1%) and 15 (2%) reported thoughts of self-harm 8–12 and 13–14 days, respectively. The most commonly reported symptom of depression was feeling down or depressed (n = 671 (95%)) 1–7 days in the last 2 weeks.

### SLE frequency at baseline

Nearly all (n = 664, 94%) of participants reported a SLE in the previous 3 months. The mean number of stressful life events (Table 3, Fig 3) in the 3 months prior to baseline interviews was 3.5 events (SD: 2.4, range: 0–15). Most participants reported 0–2 events (n = 298, 42%) or 3–5 events (n = 285, 40%). The most frequently reported SLE was financial stress indicated by going to bed hungry in the last month, reported by 48% of participants (n = 337). Other frequently reported SLEs included relationship changes (n = 318, 45%), death of a family member or friend (n = 290, 41%), serious illness of a family member or friend (n = 274, 39%), employment related SLEs (n = 230, 33%), and a new health problem themselves (n = 214, 30%). Other less commonly reported SLEs included motor vehicle accidents, physical assault, robbery, and feeling unsafe in one's neighborhood. Importantly, this study was conducted during the SARS-CoV-2 pandemic, during which frequency of these SLEs may be higher than what we

**Table 2. Descriptive statistics for depressive symptoms at baseline.**

|  |  | N (%) | Mean (SD; Range) |
|---|---|---|---|
| **PHQ-9 Score** |  | 708 | 9.4 (3.9; 5–25) |
|  | Mild depressive symptoms | 448 (63.3) |  |
|  | Moderate depressive symptoms | 181 (25.6) |  |
|  | Moderate-severe depressive symptoms | 56 (7.9) |  |
|  | Severe depressive symptoms | 19 (2.7) |  |
|  | Missing | 4 (0.6) |  |
| **Feeling down/depressed** |  |  |  |
|  | 0 days | 34 (4.8) |  |
|  | 1–7 days | 529 (74.7) |  |
|  | 8–12 days | 52 (7.3) |  |
|  | 13 or 14 days | 90 (12.7) |  |
|  | Missing | 3 (0.4) |  |
| **Little interest/pleasure** | 0 days | 104 (14.7) |  |
|  | 1–7 days | 477 (67.4) |  |
|  | 8–12 days | 52 (7.3) |  |
|  | 13 or 14 days | 64 (9.0) |  |
|  | Missing | 11 (1.6) |  |
| **Sleep issues** | 0 days | 68 (9.6) |  |
|  | 1–7 days | 425 (60.0) |  |
|  | 8–12 days | 80 (11.3) |  |
|  | 13 or 14 days | 133 (18.8) |  |
|  | Missing | 2 (0.3) |  |
| **Lethargy/fatigue** | 0 days | 48 (6.8) |  |
|  | 1–7 days | 458 (64.7) |  |
|  | 8–12 days | 86 (12.2) |  |
|  | 13 or 14 days | 113 (16.0) |  |
|  | Missing | 3 (0.4) |  |
| **Poor appetite/overeating** | 0 days | 162 (22.9) |  |
|  | 1–7 days | 398 (56.2) |  |
|  | 8–12 days | 68 (9.6) |  |
|  | 13 or 14 days | 79 (11.2) |  |
|  | Missing | 1 (0.1) |  |
| **Poor self esteem** | 0 days | 104 (14.7) |  |
|  | 1–7 days | 387 (54.7) |  |
|  | 8–12 days | 83 (11.7) |  |
|  | 13 or 14 days | 129 (18.2) |  |
|  | Missing | 5 (0.7) |  |
| **Trouble concentrating** | 0 days | 245 (34.6) |  |
|  | 1–7 days | 350 (49.4) |  |
|  | 8–12 days | 48 (6.8) |  |
|  | 13 or 14 days | 63 (8.9) |  |
|  | Missing | 2 (0.3) |  |
| **Acting slowly or fidgety** | 0 days | 350 (49.4) |  |
|  | 1–7 days | 288 (40.7) |  |
|  | 8–12 days | 26 (3.7) |  |
|  | 13 or 14 days | 42 (5.9) |  |
|  | Missing | 2 (0.3) |  |

(*Continued*)

**Table 2.** (Continued)

|  |  | N (%) | Mean (SD; Range) |
|---|---|---|---|
| **Thoughts of self-harm** | 0 days | 578 (81.6) |  |
|  | 1–7 days | 105 (14.8) |  |
|  | 8–12 days | 8 (1.1) |  |
|  | 13 or 14 days | 15 (2.1) |  |
|  | Missing | 2 (0.3) |  |

might expect in non-pandemic times, including death or serious illness of family members or friends. Interestingly, hospitalization of oneself was reported less frequently compared to serious illness or death of a family member or friend during the pandemic period.

### Bivariate analyses of recent SLEs and depressive severity at baseline

When considered individually, virtually all SLEs, with the exception of involvement in a recent motor vehicle accident, were associated with increased depressive severity among participants

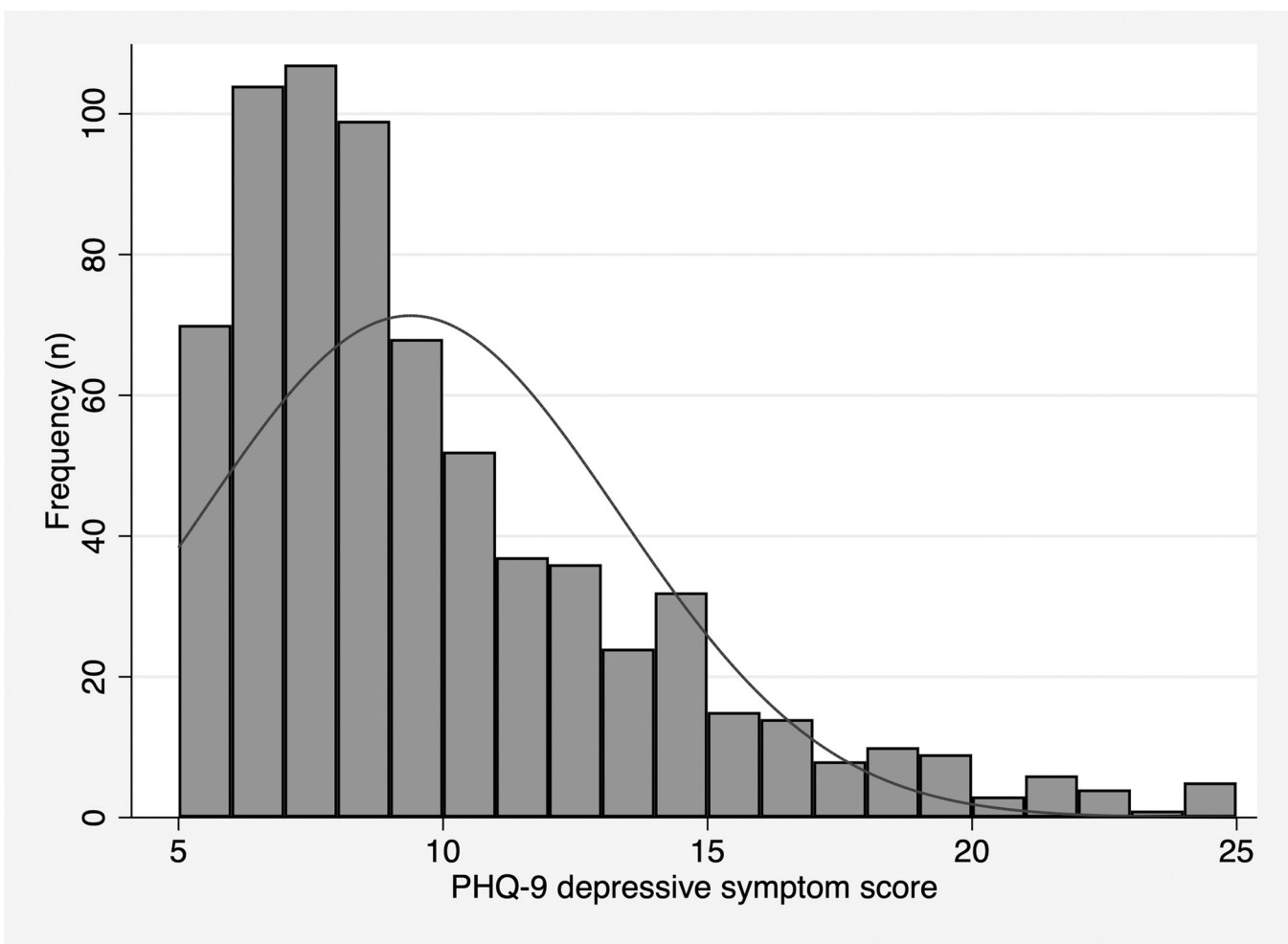

**Fig 2. Depressive severity at baseline.**

**Table 3. Descriptive statistics for stressful life events at baseline.**

| | N (%) | Missing | Mean (SD; Range) |
|---|---|---|---|
| | 708 | | |
| **Number of stressful life events** | | 0 (0.0) | 3.5 (2.4; 0–15) |
| 0–2 | 298 (42.1) | | |
| 3–5 | 285 (40.3) | | |
| 6–8 | 100 (14.1) | | |
| 9–11 | 21 (3.0) | | |
| 12–15 | 4 (0.6) | | |
| **Relationship changes/difficulties** | 318 (44.9) | 3 (0.4) | |
| Got engaged/married | 29 (4.1) | 0 (0.0) | |
| Increase in arguments with partner | 164 (23.2) | 2 (0.3) | |
| Got divorced/separated | 35 (4.9) | 1 (0.1) | |
| Estranged from family member | 209 (29.5) | 1 (0.1) | |
| **Death of family member or close friend** | 290 (41.0) | 4 (0.6) | |
| **Serious illness of family member or close friend** | 274 (38.7) | 4 (0.6) | |
| **Employment-related difficulties** | 230 (32.5) | 1 (0.1) | |
| Unable to find work | 137 (19.4) | 1 (0.1) | |
| Losing source of income | 134 (18.9) | 0 (0.0) | |
| Major problems at job | 42 (5.9) | 0 (0.0) | |
| **Major new health problem** | 214 (30.2) | 2 (0.3) | |
| **Hospitalized** | 68 (9.6) | 0 (0.0) | |
| **Motor vehicle accident** | 16 (2.3) | 0 (0.0) | |
| **Physically assaulted** | 67 (9.5) | 1 (0.1) | |
| **Robbed** | 109 (15.4) | 0 (0.0) | |
| **Felt unsafe in neighborhood** | 145 (20.5) | 0 (0.0) | |
| **Going to bed hungry in the last month** | 337 (47.6) | 0 (0.0) | 1.9 (3.1; 0–29) |
| 0 times | 371 (52.4) | | |
| 1–7 times | 311 (43.9) | | |
| 8+ times | 26 (3.7) | | |

at baseline, with prevalence ratios ranging from 1.12–1.59, although statistical significance varied (Table 4). The strongest associations were observed for having experienced a major new health problem (PR = 1.59, 95% CI: [1.31–1.93]), divorce or separation (1.54, [1.11, 2.12]), hospitalization (1.34, [1.02–1.76]), getting engaged or married (1.34, [0.91, 1.99]), being physically assaulted (1.31, [0.99, 1.73]), estrangement from a family member (1.28, [1.05–1.56]), and losing one's source of income (1.27, [1.02, 1.59]). The one association that was below the null, although its 95% CI spanned the null, was having experienced a motor vehicle accident, one of the least common events (0.86, [0.41, 1.78]).

## Multivariate analysis of recent SLEs and PHQ-9 score at baseline

When SLEs were considered as a single summary score of the number of types of events experienced in the 3 months prior to baseline, the use of a simple linear term (as opposed to a quadratic or other more flexible representation) was supported by non-significance of the quadratic term at an alpha level of 0.05 and by visualization using a LOWESS curve. In both unadjusted and adjusted models, the number of SLEs was associated with a higher PHQ-9 score at baseline (Fig 4, Table 4). In the adjusted model, each additional SLE was associated with a 9% relative increase in the risk of having moderate to severe depressive symptoms (1.09,

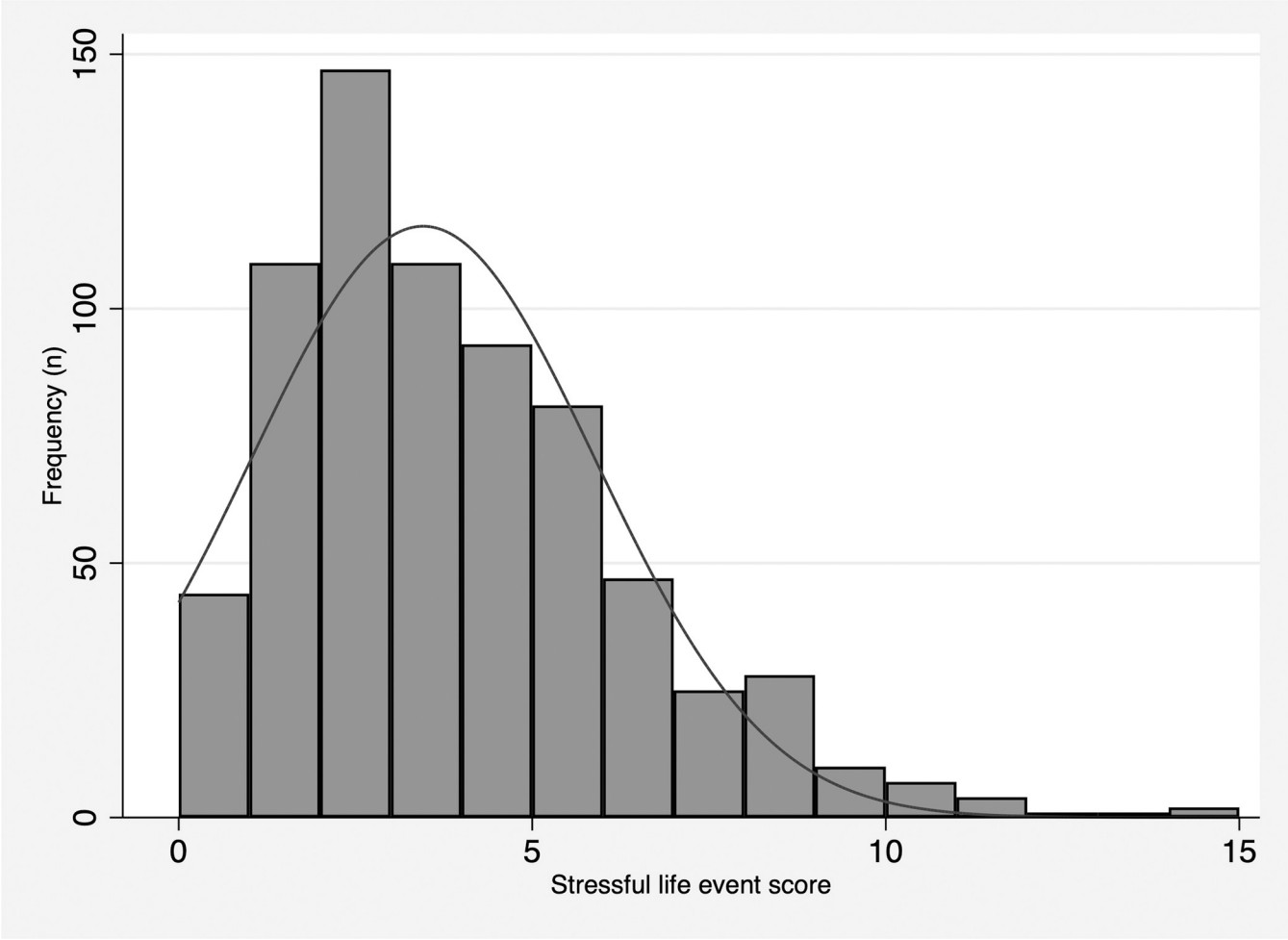

**Fig 3. Recent stressful live event frequency at baseline.**

[1.05, 1.13], p<0.001). The estimated, adjusted PR is compatible with a range of true PRs from 1.05 to 1.13.

## Discussion

In this study of adult participants with hypertension and/or diabetes and depression at baseline, we found that recent SLEs were strongly predictive of greater depressive severity. Almost every type of SLE was individually associated with greater depressive severity. Overall, a dose-response relationship was observed, with each additional SLE being associated with a 9% increase in risk of move severe depression.

Stressful events are known risk factors for depressive episodes and increased depressive severity, yet remain under researched as possible intervention points for depressive treatment among patients with NCDs in low-resource settings [36, 51–54]. Most participants had mild depressive symptoms and nearly all participants experienced at least one SLE in the 3 months prior to baseline interviews. Among the most frequent SLEs experienced were SLEs frequently reported in prior studies: relationship difficulties, recent death of a family member or close friend, serious illness of a family member or close friend, financial stress as measured by going to bed hungry, and a major new health problem [55–57].

**Table 4. Bivariate and multivariate analyses of stressful life events and depressive severity.**

| | Moderate or more severe depressive status | |
|---|---|---|
| **Stressful life events** | **PR (95% CI)*** | **p (alpha = 0.05)** |
| Relationship difficulties | 1.20 (0.99, 1.46) | 0.065 |
| Got engaged/married | 1.34 (0.91, 1.99) | 0.135 |
| Increase in arguments with partner | 1.15 (0.93, 1.44) | 0.197 |
| Got divorced/separated | 1.54 (1.11, 2.12) | 0.009 |
| Estranged from family member | 1.28 (1.05, 1.56) | 0.015 |
| Death of family member or close friend | 1.22 (1.00, 1.48) | 0.046 |
| Serious illness of family member or close friend | 1.12 (0.92, 1.37) | 0.245 |
| Employment-related difficulties | 1.16 (0.95, 1.42) | 0.139 |
| Unable to find work | 1.14 (0.91, 1.44) | 0.260 |
| Losing source of income | 1.27 (1.02, 1.59) | 0.032 |
| Major problems at job | 1.12 (0.77, 1.64) | 0.556 |
| Major new health problem | 1.59 (1.31, 1.93) | <0.0001 |
| Hospitalized | 1.34 (1.02, 1.76) | 0.038 |
| Motor vehicle accident | 0.86 (0.41, 1.78) | 0.679 |
| Physically assaulted | 1.31 (0.99, 1.73) | 0.059 |
| Robbed | 1.15 (0.89, 1.47) | 0.291 |
| Felt unsafe in neighborhood | 1.23 (0.98, 1.53) | 0.070 |
| Went to bed hungry in the last month* | 1.17 (0.96, 1.42) | 0.124 |
| **Multivariate analyses** | **PRR (95% CI)*** | **p (alpha = 0.05)** |
| Unadjusted model | 1.09 (1.05, 1.12) | <0.0001 |
| Adjusted model | 1.09 (1.05, 1.13) | <0.0001 |

*Binary: 0 days versus any days

**Prevalence ratio (PR)

***Prevalence risk ratio (PRR)

Common SLEs such as divorce/separation, estrangement from a family member, death of a family member or close friend, losing source of income, major new health problems, and hospitalization had bivariate associations for increased risk of moderate or more severe depression at baseline. Experiencing divorce/separation or having a major new health problem resulted in particularly high risk of worsened depression. Results from this study suggest that routine assessment and quantification of recent SLEs may be an important intervention point for patients with depression seeking diabetes and hypertension care at NCD clinics in similar contexts. Of note, the second most predictive SLE was major new health problems [17, 58–61]. Each of these stressors may have a unique impact on each participant which may vary over time. As well, participants may or may not routinely disclose such events to primary care practitioners, and SLEs may affect health service utilization [62, 63].

More consistent with stress sensitization hypotheses, results suggest a linear relationship between number of SLEs experienced and risk of worsened depressive severity [34–36, 51, 64]. These results further suggest that continued, formal assessment of both type and frequency of SLEs may be an opportunity to improve depressive symptom management among patients with hypertension and/or diabetes mellitus in primary, NCD care settings in this study context.

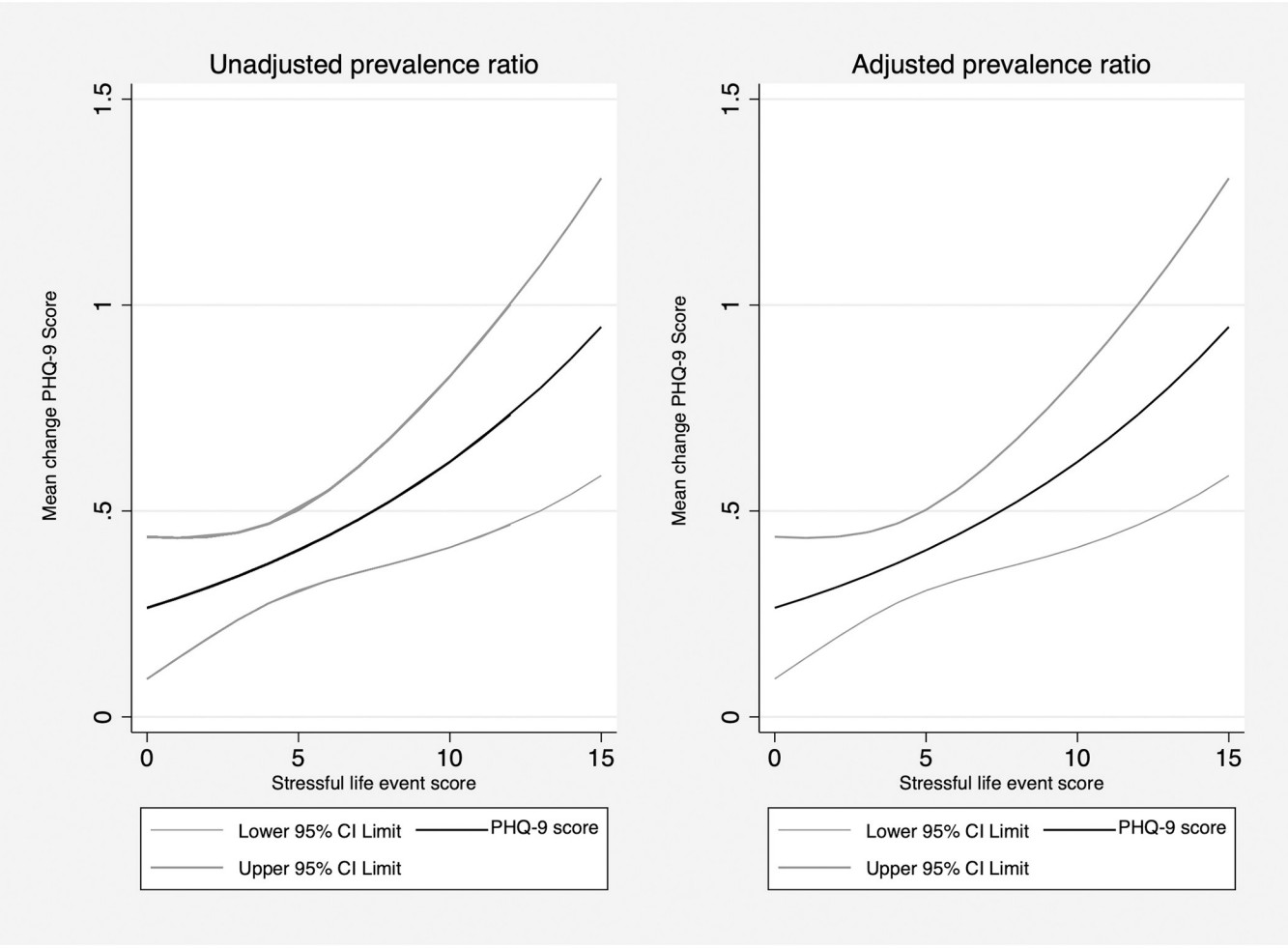

**Fig 4. Stressful life events and depressive severity at baseline: Unadjusted and adjusted prevalence ratio.**

### Strengths and limitations

The study sample is drawn from 10 geographically dispersed health facilities across Malawi, with participants identified through routine screening for depression during NCD care. The measure of depression used is robust and previously validated in Malawi [39]. Further, the measure of SLEs is widely used [43–45]. Missing data were minimal for exposure, outcome, and covariates.

Use of a continuous predictor variable assumes that all one-unit changes in SLE exposure have equal risk on moderate or worse depressive severity when some SLEs may, in reality, be stronger predictors of depressive severity than others. These effects are not constant between participants nor over time within participants, and chronicity versus acuity of events may differ by event type (e.g., experiencing IPV versus a robbery). Further, some events might rest on the same continuum of stressing processes (e.g., increase in arguments with a partner and divorce/separation). However, different types of stressors are measured separately as they may impact depression outcomes differently and on different timelines. Participants may feel social pressure to hide depressive symptoms and substance use, possibly resulting in participant bias despite few missing data existing for these covariates. This study spanned the start of the SARS-CoV-2 pandemic, during which MNS disorder prevalence, particularly for depression,

increased globally and current study procedures were modified to occur virtually rather than in-person [65, 66]. Participants could have increased MNS symptoms related to the pandemic, and could have also experienced increased SLEs related to the pandemic.

Selection bias is possible, as individuals who seek and obtain healthcare at participating clinics may differ on sociodemographic, exposure, and outcome related variables compared to participants who do not or are unable to seek care. Specifically, those with severe depression and or recent SLEs may be less likely to seek care and be selected into the study compared to participants in the study. Most participants were female, parous, and aged over 50 years, limiting generalizability of results beyond the study population and context. However, the study sample is representative of the population accessing NCD care at the study's 10 sites across Malawi in the time leading up to and including the SARS-CoV-2 pandemic.

## Conclusions

Depressive symptom management in NCD clinics in low-resource settings may benefit from routine screening for type and frequency of recent stressful events, as participants may or may not disclose occurrence of these events during routine NCD clinic visits. Targeting of limited resources to high-risk individuals benefits from improved understanding of SLEs impacting depression severity among patients with NCDs. These SLEs are important predictors of patient depression outcomes and results from this study highlight the importance of noting SLEs as possible predictors of moderate or more severe depressive severity. Future research directions include assessing the time-varying and longitudinal impact of SLEs on depressive severity over the course of enhanced and standard of care depression treatment programs.

## Supporting information

**S1 File. PHQ-9 survey.**
(PDF)

**S2 File. Stressful life events survey.**
(PDF)

**S3 File. Inclusivity in global research.**
(DOCX)

## Acknowledgments

We greatly appreciate the time and efforts of all of the study participants and study staff.

## Author Contributions

**Conceptualization:** Kelsey R. Landrum, Brian W. Pence.

**Data curation:** Kelsey R. Landrum, Brian W. Pence, Bradley N. Gaynes, Josée M. Dussault, Mina C. Hosseinipour, Kazione Kulisewa, Jullita Kenela Malava, Jones Masiye, Harriet Akello, Michael Udedi, Chifundo C. Zimba.

**Formal analysis:** Kelsey R. Landrum, Brian W. Pence.

**Funding acquisition:** Kelsey R. Landrum, Brian W. Pence.

**Investigation:** Brian W. Pence, Bradley N. Gaynes, Josée M. Dussault, Mina C. Hosseinipour, Kazione Kulisewa, Jullita Kenela Malava, Jones Masiye, Harriet Akello, Michael Udedi, Chifundo C. Zimba.

**Methodology:** Kelsey R. Landrum, Brian W. Pence.

**Project administration:** Kelsey R. Landrum.

**Supervision:** Brian W. Pence, Bradley N. Gaynes.

**Visualization:** Kelsey R. Landrum.

**Writing – original draft:** Kelsey R. Landrum, Brian W. Pence.

**Writing – review & editing:** Kelsey R. Landrum, Brian W. Pence, Bradley N. Gaynes, Josée M. Dussault, Mina C. Hosseinipour, Kazione Kulisewa, Jullita Kenela Malava, Jones Masiye, Harriet Akello, Michael Udedi, Chifundo C. Zimba.

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
