## [Decision Letter · Decision Letter 0]

2 Aug 2022

PONE-D-22-06008

The cross-sectional association of stressful life events with depression severity among patients with hypertension and diabetes in Malawi

PLOS ONE

Dear Dr. Landrum,

Thank you for submitting your manuscript to PLOS ONE. After careful consideration, we feel that it has merit but does not fully meet PLOS ONE’s publication criteria as it currently stands. Therefore, we invite you to submit a revised version of the manuscript that addresses the points raised during the review process.

The manuscript has been evaluated by two reviewers, and their comments are available below.

The reviewers have raised a number of concerns. They request improvements to the reporting of methodological aspects of the study (such as how the DAGs were constructed, and the description of the depression covariate), and revisions to the discussion.

Could you please carefully revise the manuscript to address all comments raised?

We look forward to receiving your revised manuscript.

Kind regards,

Lorena Verduci

Staff Editor

PLOS ONE

https://journals.plos.org/plosone/s/file?id=ba62/PLOSOne_formatting_sample_title_authors_affiliations.pdf".

“This study received funding from the National Institute of Mental Health (MPIs: Brian W. Pence, PhD, Mina C. Hosseinipour, MD MPH; Jones Masiye; U19MH113202-01). KRL received support from supplement U19MH113202-04S1. JD received support from supplement U19MH113202-01S2 and T32AI070114.  Neither the funding agency nor funders had any part in study conceptualization, data collection, data analysis, or decision-making about publication.”

“This study received funding from the National Institute of Mental Health (MPIs: Brian W. Pence, PhD, Mina C. Hosseinipour, MD MPH; Jones Masiye; U19MH113202-01). KRL received support from supplement U19MH113202-04S1. JD received support from supplement U19MH113202-01S2 and T32AI070114.  Neither the funding agency nor funders had any part in study conceptualization, data collection, data analysis, or decision-making about publication.”

Upon re-submitting your revised manuscript, please upload your study’s minimal underlying data set as either Supporting Information files or to a stable, public repository and include the relevant URLs, DOIs, or accession numbers within your revised cover letter. For a list of acceptable repositories, please see

http://journals.plos.org/plosone/s/data-availability#loc-recommended-repositories. Any potentially identifying patient information must be fully anonymized.

Reviewers' comments:

Reviewer's Responses to Questions

**Comments to the Author**

1. Is the manuscript technically sound, and do the data support the conclusions?

Reviewer #1: Partly

Reviewer #2: Yes

2. Has the statistical analysis been performed appropriately and rigorously? 

Reviewer #1: I Don't Know

Reviewer #2: Yes

3. Have the authors made all data underlying the findings in their manuscript fully available?

Reviewer #1: Yes

Reviewer #2: Yes

4. Is the manuscript presented in an intelligible fashion and written in standard English?

Reviewer #1: Yes

Reviewer #2: Yes

5. Review Comments to the Author

Reviewer #1: It is an interesting and welcome paper on adults with NCDs. This population should require more attention in LMICs, especially when they also suffer from depressive symptoms. The paper is well structured and references are numerous.

Though I have some questions and comments:

I would not say that depression remains “disproportionately “ high in LMICs seeing the high diversity in prevalence rates that are given in the following rates. These rates should be a bit more contextualized as you may have different screening tools with various identified period (from the last 2 weeks to in the lifecourse) and different sampled population.

Qualifying SLEs as "promising" risk factors seems to me a bit clumsy. It also rises the question of the objective of the paper. Do the authors want to promote SLEs as a predictors of depression? are they really easier to identify than depression in a data collection? Looking at the SF1 & 2, it does not look like but perhaps it is because of the sensitivity of the questions related to depression? in this case, it has to be clearly mentioned.

I am also wondering how the interaction Depression*SLE*NCD is treated in the paper. To me, it looks like this study could be conducted on general population which is not. I am curious to know how many people with NCDs do not show any depressive symptoms for instance to understand how large is the combination NCD*depression. That would support the choice that has been done here to conduct the study only on people who show some depressive signs. Otherwise, this selection is questioning and should be discussed. Indeed, we could also think that many people, on the contrary do not develop depressive symptoms while they have experiences SLEs.

The data collection occurred during the covid pandemic: Has the covid-pandemic a dramatic impact in Malawi? Not sure it has, however, it may have impacted life conditions at least for a certain period of time. It could also change the way people were interviewed with the need to respect some physical distance. It could be helpful to develop a bit more for the readers who do not know the context.

I believe the analysis is well done and statistically appropriate but I am questioning about the relevance to estimate a constant cumulative risk when having experienced an additional SLE with some that have been shown to impact more than others.

I was also surprised not to see in the discussion about the timeline of the SLEs. Indeed, except for injuries (that are in fact insignificant), many events result from processes: being diagnosed from a new disease result from having symptoms enough worrying to consult, getting married or divorced result from previous preliminary events and depression could become worsen on the long run.

It is also true for physical assault that may be a repetitive event for some participants. It is quite high for this population and it would have been welcome to test some specific gender-oriented associations between SLE and depression. Again, I was wondering what would have been the prevalence of depression in a population who experience so high insecurity feelings and if people with NCDs are specific from this respect.

Finally, to come back to one of my first remark, there is a need to more connect these results to the fact that the participants live with NCDs. I have two suggestions to do so: first is to test the association between depression severity and the specific NCD the participants suffer from and including it in the models. The second is to explore the association between NCDs and the SLE: it is probably not independent to be sick and lose a job or a spouse. At least, all these issues should be discussed.

Punctual remarks

The description of the depression covariate should clearly mention that the screening collect symptoms for the last 2 weeks at the baseline.

The DAG presented in SF3 should be included in the paper.

The frequencies of the people excluded because of missing data and of disagreement in the screening should be reported in parenthesis.

thresholds are for some of them a bit surprising in tables.

the authors should make a distinction between having no experience and having at least 1 or 2 as the firsts are supposed to be very different from the others, and group the last 2 groups (9+) that are really a minority.

going to bed hungry: group the last 3 groups that are really a minority.

there is no need to qualify the parity, just mention the number of children.

Reviewer #2: The authors conduct a cross-sectional study to estimate the association between SLEs and depressive severity among 708 patients who seek for NCD care. The results showed that SLEs in the prior 3 months were associated with greater depressive severity.

1. Confounders adjusted in the final model were identified through the DAGs. Please explain how the DAGs were constructed.

2. Participants were recruited across 10 study sites. Are the participants from different sites homogeneous?

6. PLOS authors have the option to publish the peer review history of their article (what does this mean?). If published, this will include your full peer review and any attached files.

Reviewer #1: **Yes: **Géraldine Duthé

Reviewer #2: No

---

## [Author Response · Author response to Decision Letter 0]

13 Oct 2022

5 September 2022

RE: “The cross-sectional association of stressful life events with depression severity among patients with hypertension and diabetes in Malawi”

Dear Dr. Duthé and Reviewer 2,

Thank you very much for your thoughtful comments on our manuscript entitled “The cross-sectional association of stressful life events with depression severity among patients with hypertension and diabetes in Malawi”. These comments and suggestions have been helpful in further revising our manuscript. We have responded to each of your comments in line below, and accordingly in the tracked changes of the manuscript. We hope that these changes adequately address your questions.

Sincerely,

Kelsey R. Landrum

Reviewer #1: It is an interesting and welcome paper on adults with NCDs. This population should require more attention in LMICs, especially when they also suffer from depressive symptoms. The paper is well structured and references are numerous.

Thank you for your thoughtful comments regarding the study topic and overall structure. 

Though I have some questions and comments:

I would not say that depression remains “disproportionately “ high in LMICs seeing the high diversity in prevalence rates that are given in the following rates. These rates should be a bit more contextualized as you may have different screening tools with various identified period (from the last 2 weeks to in the lifecourse) and different sampled population.

Thank you for this great point. We have elaborated on the range of prevalence values, and study differences that might lead to such a range, including varying depression screening tools, study populations, outcome ascertainment timelines, and study settings.

Qualifying SLEs as "promising" risk factors seems to me a bit clumsy. It also rises the question of the objective of the paper. Do the authors want to promote SLEs as a predictors of depression? are they really easier to identify than depression in a data collection? Looking at the SF1 & 2, it does not look like but perhaps it is because of the sensitivity of the questions related to depression? in this case, it has to be clearly mentioned.

We have clarified our language to represent our original intent of this sentence, to state that SLEs are possible risk factors for identifying patients at risk of depressive episodes and possible intervention points for depression care. 

I am also wondering how the interaction Depression*SLE*NCD is treated in the paper. To me, it looks like this study could be conducted on general population which is not. I am curious to know how many people with NCDs do not show any depressive symptoms for instance to understand how large is the combination NCD*depression. That would support the choice that has been done here to conduct the study only on people who show some depressive signs. Otherwise, this selection is questioning and should be discussed. Indeed, we could also think that many people, on the contrary do not develop depressive symptoms while they have experiences SLEs.

Thank you for these insightful comments. The present study is situated within a larger, longitudinal clinic randomized control trial. The target population of both the parent study and the present study is adults accessing NCD care treatment (specifically hypertension and/or diabetes care). While not all individuals living with NCDs will go on to develop depression or another MNS disorder, depression has been previously associated with hypertension, diabetes, asthma, and other NCD diagnoses. The comorbidity of depression and NCDs is complex, but existing NCD infrastructure may be able to be leveraged to improve CMD care access. The interaction between all three (SLEs, depression, and a given NCD) has yet to be established and agreed upon in the literature, particularly for Malawian populations. This study lays the groundwork for exploring this topic further. 

The data collection occurred during the covid pandemic: Has the covid-pandemic a dramatic impact in Malawi? Not sure it has, however, it may have impacted life conditions at least for a certain period of time. It could also change the way people were interviewed with the need to respect some physical distance. It could be helpful to develop a bit more for the readers who do not know the context.

Thank you for noting this. We agree with this point and have expanded our discussion of the pandemic into the introduction, results, and discussion sections accordingly.

I believe the analysis is well done and statistically appropriate but I am questioning about the relevance to estimate a constant cumulative risk when having experienced an additional SLE with some that have been shown to impact more than others.

Thank you for raising this point. We agree that each SLE may not have an equal effect on depression, as noted in our “Strengths and Limitations” section: “Use of a continuous predictor variable assumes that all one-unit changes in SLE exposure have equal risk on moderate or worse depressive severity when some SLEs may, in reality, be stronger predictors of depressive severity than others. These effects are not constant between participants nor over time within participants.” For instance, experiencing a death of a family member may affect an individual’s depression outcome differently compared to feeling unsafe in one’s neighborhood. Given that the Life Experiences Survey used in this study asks about presence or absence of each stressful life event, and does not ask participants to quantify the impact of each such event on various outcomes, we estimate the prevalence ratio for having moderate or more severe depressive severity per additional SLE.1,2 This approach is consistent with a number of other studies in this area that have similarly examined a continuous variable of the number of stressful life events as a predictor of a range of health outcomes.3,4 To your point, we also present the prevalence ratio of depression per each SLE (e.g., got engaged/married, major problems at job, major new health problem, etc.) in the bivariate analysis of SLEs and depressive severity in Table 4. 

I was also surprised not to see in the discussion about the timeline of the SLEs. Indeed, except for injuries (that are in fact insignificant), many events result from processes: being diagnosed from a new disease result from having symptoms enough worrying to consult, getting married or divorced result from previous preliminary events and depression could become worsen on the long run.

Thank you for raising this important point. The timeline of SLEs may differ for each person and for each type of SLE, as you exemplify above. The chronicity versus acuity of events might differ as you also point out.5–12 The original Life Experiences Survey tool asks about presence or absence of each SLE in the past zero to six months or seven months-one year. We modified our use of the survey to ask about presence or absence of each SLE even more recently, in the past three months. Certainly experiencing, for example, an increase in arguments with a partner AND getting divorced/separated may be part of a continuum of a process. However, each are measured as separate SLEs and both may impact depression outcomes differently. We have elaborated on the above in the “Discussion” section, which we believe effectively addresses these important points.

It is also true for physical assault that may be a repetitive event for some participants. It is quite high for this population and it would have been welcome to test some specific gender-oriented associations between SLE and depression. Again, I was wondering what would have been the prevalence of depression in a population who experience so high insecurity feelings and if people with NCDs are specific from this respect.

Thank you also for this great comment. Multiple of these events, such as intimate partner violence as you discuss, may occur chronically and repetitively. We did consider stratifying our analyses by gender, noting that gender is likely an important confounder, as it may increase the risk of experiencing certain SLEs (e.g., robbery, assault, feeling unsafe in neighborhood, major problem at job, etc.) and depression. Nearly 80% of participants are female, and approximately 20% male, limiting our power to detect differences in SLEs on depression outcomes among male participants compared to female participants. 

Finally, to come back to one of my first remark, there is a need to more connect these results to the fact that the participants live with NCDs. I have two suggestions to do so: first is to test the association between depression severity and the specific NCD the participants suffer from and including it in the models. The second is to explore the association between NCDs and the SLE: it is probably not independent to be sick and lose a job or a spouse. At least, all these issues should be discussed.

Thank you also for this important comment. NCDs and depression are frequently comorbid and you raise a great point that the NCDs themselves might also be related to experiencing an SLE.13–18 All participants have either hypertension, diabetes, or both. We have clarified these relationships in the manuscript accordingly. 

Punctual remarks

The description of the depression covariate should clearly mention that the screening collect symptoms for the last 2 weeks at the baseline. 

We have further clarified this, thank you.

The DAG presented in SF3 should be included in the paper.

We have made this change. The DAG is now presented in the “Methods” section as Figure 1.

The frequencies of the people excluded because of missing data and of disagreement in the screening should be reported in parenthesis.

We have reported the number of missing and excluded observations.

thresholds are for some of them a bit surprising in tables.

We are unsure about what thresholds and tables are being referred to here, but have carefully reviewed tables and thresholds for appropriateness and to confirm they are supported by the given citations. 

the authors should make a distinction between having no experience and having at least 1 or 2 as the firsts are supposed to be very different from the others, and group the last 2 groups (9+) that are really a minority.

Thank you for raising this point. There are fewer participants with 9-11 and >12 SLEs. We specification used was the best choice after examining linearity. Quadratic, linear, categorical, and binary (dichotomized at the median with the 90th-100th percentiles restricted to the median values of those values) were considered and the current categorization was the best choice. AICs and LRTs were used to compare functional forms in nested and AICs to compare non-nested models. We also conducted a separate analysis with categorization of the exposure variable collapsing the last 2 groups, noting that precision did not change greatly between the this categorization and the others.

going to bed hungry: group the last 3 groups that are really a minority.

We have collapsed the last three strata accordingly (Table 3).

there is no need to qualify the parity, just mention the number of children.

We have modified this accordingly (Table 1).

Reviewer #2: The authors conduct a cross-sectional study to estimate the association between SLEs and depressive severity among 708 patients who seek for NCD care. The results showed that SLEs in the prior 3 months were associated with greater depressive severity.

1. Confounders adjusted in the final model were identified through the DAGs. Please explain how the DAGs were constructed.

Thank you very much for your comments. Our DAG (now Figure 1) was constructed according to standard DAG protocol as explained by Hernán and Robins (2020).19 The DAG is based on knowledge in current literature and subject matter expertise. Social support, PTSD, and Anxiety can be affected by SLEs and mediate their impact on depression remission outcomes.20,21 Women may be more likely to experience SLEs (e.g., IPV) and to be at higher risk of depression.5,10 Age affects the number of possible SLEs experienced, as well as risk of depression.5,8 Education is highly correlated with parity and age in many settings, and both impact risk of experiencing the SLEs we measure and achievement of depression remission.5,22 

2. Participants were recruited across 10 study sites. Are the participants from different sites homogeneous?

Thank you for raising this great point. Participants across the 10 study sites are geographically representative of all major geographic regions in Malawi and enrollment frequency was similar across all study site. We have added participant enrollment by study site to Table 1 and clarified that participants across all study sites were similar according to Table 1 characteristics.

 

References

1. Sarason, I. G. & Johnson, J. H. The Life Experiences Survey: Preliminary Findings. https://apps.dtic.mil/sti/citations/ADA027527 (1976).

2. Sarason, I. G., Johnson, J. H. & Siegel, J. M. Assessing the impact of life changes: Development of the Life Experiences Survey. Journal of Consulting and Clinical Psychology 46, 932–946 (1978).

3. Mugavero, M. J. et al. Overload: impact of incident stressful events on antiretroviral medication adherence and virologic failure in a longitudinal, multisite human immunodeficiency virus cohort study. Psychosom Med 71, 920–926 (2009).

4. Reif, S. et al. Highly Stressed: Stressful and Traumatic Experiences among individuals with HIV/AIDS in the Deep South. AIDS Care 23, 152–162 (2011).

5. Lund, C. et al. Social determinants of mental disorders and the Sustainable Development Goals: a systematic review of reviews. The Lancet Psychiatry 5, 357–369 (2018).

6. Amital, D., Fostick, L., Silberman, A., Beckman, M. & Spivak, B. Serious life events among resistant and non-resistant MDD patients. Journal of Affective Disorders 110, 260–264 (2008).

7. Dohrenwend, B. P. Inventorying Stressful Life Events as Risk Factors for Psychopathology: Toward Resolution of the Problem of Intracategory Variability. Psychol Bull 132, 477–495 (2006).

8. Allen, J., Balfour, R., Bell, R. & Marmot, M. Social determinants of mental health. International Review of Psychiatry 26, 392–407 (2014).

9. Bridges, S. & Disney, R. Debt and depression. Journal of Health Economics 29, 388–403 (2010).

10. Howard, L. M., Oram, S., Galley, H., Trevillion Kylee & Feder, G. Domestic violence and perinatal mental disorders: a systematic review and meta-analysis. PLoS Medicine 10, (2013).

11. Lund, C. et al. Poverty and common mental disorders in low and middle income countries: A systematic review. Social Science & Medicine 71, 517–528 (2010).

12. Turunen, E. & Hiilamo, H. Health effects of indebtedness: a systematic review. BMC Public Health 14, (2014).

13. Liu, Q. et al. Changes in the global burden of depression from 1990 to 2017: Findings from the Global Burden of Disease study. Journal of Psychiatric Research 126, 134–140 (2020).

14. Chapman, D. P., Perry, G. S. & Strine, T. W. The vital link between chronic disease and depressive disorders. Prev Chronic Dis 2, A14 (2005).

15. Wulsin, L. R. & Singal, B. M. Do depressive symptoms increase the risk for the onset of coronary cisease? A systematic quantitative review. Psychosomatic Medicine 65, 201–210 (2003).

16. Mancuso, C. A., Peterson, M. G. E. & Charlson, M. E. Effects of depressive symptoms on health-related quality of life in asthma patients. J Gen Intern Med 15, 301–310 (2000).

17. Rugulies, R. Depression as a predictor for coronary heart disease: a review and meta-analysis1 1The full text of this article is available via AJPM Online at www.ajpm-online.net. American Journal of Preventive Medicine 23, 51–61 (2002).

18. Sotosky, J. R., McGrory, C. H., Metzger, D. S. & DeHoratius, R. J. Arthritis problem indicator: preliminary report on a new tool for use in the primary care setting. Arthritis Care Res 5, 157–162 (1992).

19. Miguel A., H. & James M., R. Causal Inference: What If, Chapter 6: Graphical representation of causal effects. in Causal Inference: What If 69–80 (Chapman & Hall/CRC, 2020).

20. Weinberg, A., Kujawa, A. & Riesel, A. Understanding Trajectories to Anxiety and Depression: Neural Responses to Errors and Rewards as Indices of Susceptibility to Stressful Life Events. Curr Dir Psychol Sci 31, 115–123 (2022).

21. Rane, M. S. et al. Depression and Anxiety as Risk Factors for Delayed Care-Seeking Behavior in Human Immunodeficiency Virus–Infected Individuals in South Africa. Clinical Infectious Diseases 67, 1411–1418 (2018).

22. Iwata, H. et al. Prevalence of postpartum depressive symptoms during the first 6 months postpartum: Association with maternal age and parity. Journal of Affective Disorders 203, 227–232 (2016).

---

## [Decision Letter · Decision Letter 1]

12 Dec 2022

The cross-sectional association of stressful life events with depression severity among patients with hypertension and diabetes in Malawi

PONE-D-22-06008R1

Dear Dr. Landrum,

We’re pleased to inform you that your manuscript has been judged scientifically suitable for publication and will be formally accepted for publication once it meets all outstanding technical requirements.

Kind regards,

Yann Benetreau

Staff Editor

PLOS ONE

Additional Editor Comments (optional):

Reviewers' comments:

Reviewer's Responses to Questions

**Comments to the Author**

1. If the authors have adequately addressed your comments raised in a previous round of review and you feel that this manuscript is now acceptable for publication, you may indicate that here to bypass the “Comments to the Author” section, enter your conflict of interest statement in the “Confidential to Editor” section, and submit your "Accept" recommendation.

Reviewer #1: All comments have been addressed

Reviewer #2: All comments have been addressed

2. Is the manuscript technically sound, and do the data support the conclusions?

Reviewer #1: Yes

Reviewer #2: (No Response)

3. Has the statistical analysis been performed appropriately and rigorously? 

Reviewer #1: Yes

Reviewer #2: (No Response)

4. Have the authors made all data underlying the findings in their manuscript fully available?

Reviewer #1: (No Response)

Reviewer #2: (No Response)

5. Is the manuscript presented in an intelligible fashion and written in standard English?

Reviewer #1: Yes

Reviewer #2: (No Response)

6. Review Comments to the Author

Reviewer #1: (No Response)

Reviewer #2: (No Response)

7. PLOS authors have the option to publish the peer review history of their article (what does this mean?). If published, this will include your full peer review and any attached files.

Reviewer #1: **Yes: **Géraldine Duthé

Reviewer #2: No

---

## [Editor Report · Acceptance letter]

19 Dec 2022

PONE-D-22-06008R1 

The cross-sectional association of stressful life events with depression severity among patients with hypertension and diabetes in Malawi 

Dear Dr. Landrum:

I'm pleased to inform you that your manuscript has been deemed suitable for publication in PLOS ONE. Congratulations! Your manuscript is now with our production department. 

Kind regards, 

on behalf of

Dr. Yann Benetreau 

Staff Editor

PLOS ONE